# Multi-Criteria Method for Sustainable Design of Energy Conversion Systems

**Kathleen Mallard \***, **Vincent Debusschere**  **and Lauric Garbuio**

CNRS, Grenoble INP Institute of Engineering and Management, Université Grenoble Alpes, G2Elab, 38000 Grenoble, France; Vincent.Debusschere@g2elab.grenoble-inp.fr (V.D.); lauric.garbuio@grenoble-inp.fr (L.G.)
* Correspondence: kathleen.mallard@g2elab.grenoble-inp.fr

**Abstract:** Energy production systems for isolated communities lacking national energy grids are, in many countries, associated with first energy access of rural or developing regions. Those communities require innovative design methods to select relevant solutions for sustainable developments in a context of continuously strengthening climate change conditions. The design of an innovative solution goes through multiple stages. After identifying opportunities, analyzing a context and identifying a problem, we are interested here in the process of imagining solutions and guiding reflections so that the resulting solutions are sustainable. Sustainability is analyzed from technical, economic, environmental and social angles. The two main visions for imagining solutions, the value proposition and the technical solution, are discussed. We are then developing a multi-criteria method of sustainable design to imagine the technical solution of an electricity production system in a context of first access to energy for isolated communities. This method serves as decision and discussion support between all stakeholders (community, decision makers, project managers) so that they collectively build a sustainable solution. As the exchanges progress, criteria from different fields meet and complement each other to allow the development of the specifications for the energy production solution which will be ultimately developed.

**Keywords:** sustainable design; decision support tool; first energy access

## 1. Introduction

Isolated communities experience an increase in their quality of life as soon as they gain access to electricity, through significant social and economic local developments. To propose such access to energy in developing countries, the focus should be done on energy systems based on local renewable primary resources.

### 1.1. Context of the Work

Not just to list existing technical solutions, but to propose a dedicated answer to a dedicated need, a method should be developed to select a *sustainable* solution that allows an isolated community to harness local energy (for instance from hydro-power) in a context of first access to energy. The sustainability of the solution is determined by its capacity to propose acceptable performance indicators on four major criteria that rely on technical, economic, social and environmental aspects. The relevance of such adequately weighted (optimal) solution is significant for a community in the sense that favoring only one of those criteria will lead to unbalanced developements and possibly "make things worse".

Prior to the selection of the solution, defining correctly the various aspects of the four criteria of sustainable development to correctly integrate them in a design method is the first step of proposing the

optimal answer to the identified need of an isolated community. That constitute the subject discussed in this paper, focusing on possibilities of electricity production from hydraulic energy using low-power hydroelectric plants or tidal turbines for the illustration, targeting isolated areas near seas and rivers.

*1.2. Designing Energy Systems*

Designing a technical solution, adapted to an identified need, follow stages of a process clearly defined for innovative solution [1]:

- Understanding the context and looking for opportunities;
- The analysis of the need which leads to the formulation of the problem;
- The definition of the specifications of the solution or imagined solutions;
- The sketching or pre-sizing phase of the imagined solution and its evaluation according to criteria defined upstream;
- The sizing and implementation phase of the solution.

When considering the design of innovative, and possibly low-tech, energy systems, the first four stages are particularly relevant in the context of the first access to energy by renewable resources of populations living in isolated areas. Through this work, we wish to make a contribution to the local sustainable development of these populations.

In this paper, we start by analyzing the sustainability of a solution in all its forms: technical, economic, environmental and social. We then explain that there are two points of view for imagining a solution: the external vision gives the value proposition and the internal vision gives the technical description of the product designer. We then propose a tool to imagine an energy solution under the internal vision. This decision support tool guides the designer and all stakeholders of the electrification project towards a design that allows the solution to be as sustainable as possible.

## 2. Sustainability for the Design of Energy Systems

The sustainability of technical solutions for energy systems strongly depends on the definition given to these terms during the design stage. Looking for multi-criteria design criteria of an electric power generation solution, sustainability can be assessed with four main categories of questions, described in the upcoming subsections.

- *Is the solution technically efficient?* It is mainly the priority of the designer: the solution must meet the targeted technical need according to criteria which are often usage efficiency, but whose relevance may be questionable in certain configurations.
- *Is it economically accessible to users and viable for the manager?* It is the priority of users who pay for electricity, of decision-makers and investors who demand profitability, and of managers who ensure the cost of the energy produced.
- *Is it environmentally friendly?* The environment can be modeled via near or far, local or global effects. From the manufacturing to the end of life, through the use phase, the producing electrical energy device impacts many environments negatively or positively.
- *Does it socially impact the actors present in its environment?* The actors can be the manufacturers, the installation managers, the consumers of the electricity produced, and more rarely the designers. Are local people receiving social benefits such as better education or health services, or are they suffering consequences caused by the integration of the energy system? We seek to quantify an impact on social development and well-living in the vicinity of the solution.

*2.1. Technical Durability: Performance and Reliability*

The technical performance of an energy production facility is directly related to its usage definition, thus critical. The most well-known technical criteria are quantifiable: lifetime, installed power, energy produced, power factor, efficiency, electrical quantities, energy quality, etc. They are simple

to assess when the technology is known. Other criteria are more difficult to assess such as ease of manufacture, installation or reliability of the installation.

The reliability of electrical installations represent a real problem of rural electrification [2,3]. Theoretically, the reliability of a component or system is characterized by the average time between two failures $\theta_f$ (mean time between failures, MTBF). Reliability can be studied with an approach on the life cycle of components (considering five stages: introduction, maturity, aging, end of life and failure by adding a maintenance stage) [4]. If the MTBF is constant, the failure rate $\lambda$ is the inverse of reliability: $\lambda = 1/\theta_f$. When a failure occurs, the system is unavailable for the average repair time $\theta_r$ (mean time to return, MTTR). This time between failure and restarting includes the routing of operators and spare parts which can be considerably extended depending on the context, going from onshore to offshore in the case of wind power for example [5]. The availability $A$ is then calculated from the previous quantities: $A = \theta_f/(\theta_f + \theta_r)$. The goal is to maximize availability by increasing the reliability of the overall system and decreasing the repair time to increase the energy produced. To have high availability, five conditions are required according to [5]: prevention by choosing reliable systems, removal of defective components, predicting and avoiding breakdowns by working in less dangerous regimes, fault tolerance and finally repair.

### 2.2. Financial Sustainability: Cost of Energy and Accessibility

Economic criteria are commonly analyzed with the same importance as technical criteria. Indeed, the ideal being to choose the products or services offering the best technical performance at the lowest price or at least to seek the best technical and economic compromise. When several resources are available at the same place and therefore several types of renewable and non-renewable energy production are possible, investors and decision-makers need clear indicators to make their choice. They take into account access to the resource, the yields and lifetimes of production systems, the adequacy with the energy demand and the profitability of the installation.

Financial profitability is usually analyzed with the levelized cost of energy (LCOE), presented in Equation (1). Decreasing the LCOE of an energy production system means lowering investment and operational costs as well as increasing the energy produced. The investments are proportional to the power, the complexity of the installation and the yield generally. Progress on reliability impacts operational costs by reducing maintenance, which in turns impacts the energy produced by the increasing availability. Reducing the complexity of the system can improve reliability but also decrease efficiency: designers must find a compromise between complexity and efficiency in order to achieve financial profitability. Finally, the load factor can be increased by reducing the nominal power of the installation, which also has a positive effect on CAPEX.

$$\text{LCOE} = \frac{\sum_{t=1}^{n} \frac{I_t + M_t + F_t}{(1+r)^t}}{\sum_{t=1}^{n} \frac{E_t}{(1+r)^t}} \tag{1}$$

With $I_t$ the initial investment (CAPEX); $M_t$ the operation and maintenance for the year $t$ (OPEX); $F_t$ the fuel used for the year $t$; $r$ the discount rate; $E_t$ the produced energy for the year $t$ and $n$ the lifetime.

For renewable energy productions, the LCOE tool can be declined in a spatial version, the spatial LCOE. Indeed, the resource is often not uniform on the territory, the installation and the maintenance of the electric system cost more or less according to the distances. For example, a farm project can be studied upstream thanks to the spatial LCOE by taking into account the resource, the energy extracted and the cost of production (CAPEX and OPEX) at each location in the candidate area [6–8]. As an illustration, Figure 1 presents a map of an estuary that can accommodate a tidal turbine farm and the associated spatial LCOE. In the center of the map, the blue area is the one that achieves the lowest energy cost, under £0.5/kWh.

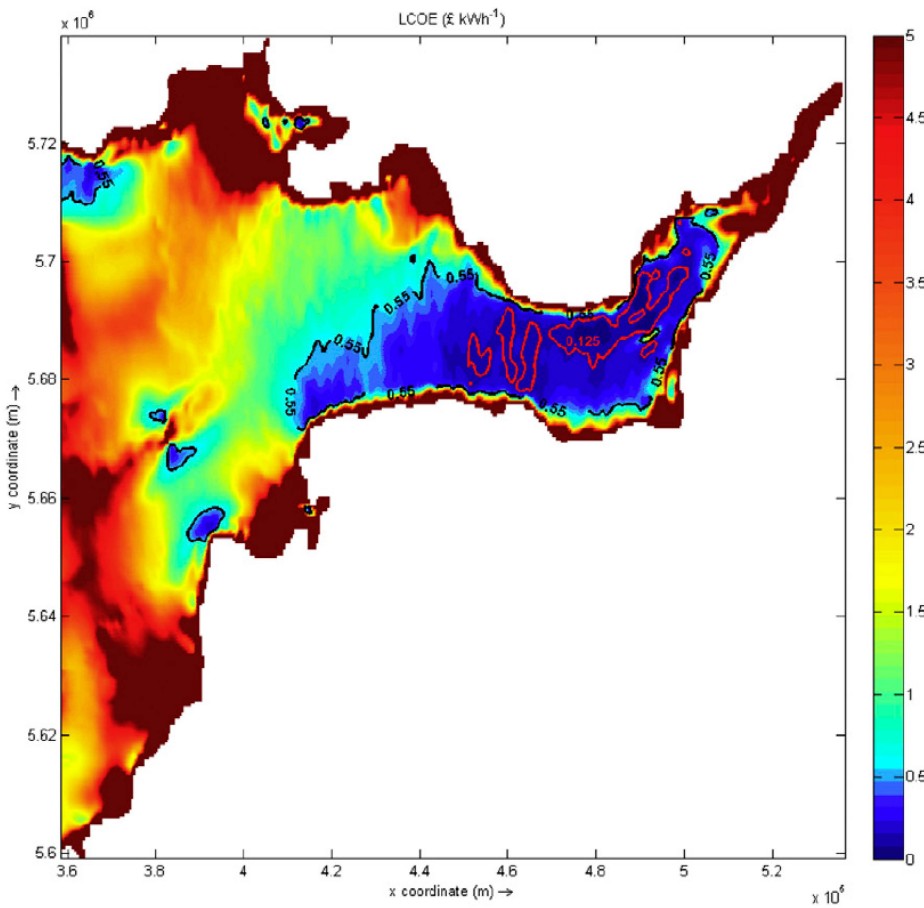

**Figure 1.** Illustration of the spatial LCOE (in £/kWh) for a tidal turbine farm in the Severn estuary near the Bristol Channel (United Kingdom) [8].

Economic sustainability is not only the profitability of the managers of the installation. The financial accessibility of the service for the beneficiaries is also important [2,3,9]. The price of electricity, the existence of subsidies or payment facilities for the service are factors which weigh heavily on household budgets [10,11]. In addition, the local economy can be positively impacted by enabling income-generating activities in homes and businesses, thus favoring job creation, or in a negative way by entering into confrontation with economic activities already established on the territory [12].

### 2.3. Environmental Sustainability: Life Cycle Analysis and Impact Study

The environmental sustainability of a product can be assessed by life cycle assessment (LCA) which is a standardized assessment method of environmental impacts over the entire lifespan (ISO 14044): "compilation and assessment of inputs, outputs and potential environmental impacts of a product system (good or service) throughout its life cycle, from the extraction of the raw material to the disposal of the product". The life cycle takes into account: manufacturing, transport and installation, operation with maintenance, uninstallation and end of life with recycling or land-filling. Overall, LCA provides indicators of the product's contribution to environmental problems such as climate change, toxicity (human and ecosystems) or the scarcity of resources.

Born in the 1960s, the LCA study presents four stages: the first and most important is the definition of the context and the functional unit (function of the product and definition of its environment to set the boundaries of the study ), then the inventory of all the elements constituting the product, the evaluation of the impacts and finally the interpretation of the results. Software were designed to move from the

inventory of all the constituent elements of the product to the associated environmental impacts, such as SimaPro, and are linked to widely supplied databases such as Ecoinvent or Idemat [13] which are valid especially for industrialized countries. The quantified impacts are the resources consumed and the emissions to air, water and soil. Figure 2, from the Innomat training [14], presents the environmental impacts of LCA in intermediate ("midpoint"), final and unique indicators which aggregate the previous ones. The analysis of the results can be conducted on specific elements (acidification, depletion of water, etc.) or by areas of protection (human health, environment or natural resources) or with a single indicator: the carbon footprint measures the impact on climate change. Eco-costs are the (financial) costs to invest to prevent the environmental impacts of the product. It should be noted that the choice of an aggregated criterion as a single indicator can lead to unreliable design results because they are very sensitive to the quality of the initial data or models. Conversely, considering only one criterion (such as the carbon footprint, for example) leads to biased results, or pollution transfers are often made without being detected, except to conduct a new LCA "verification".

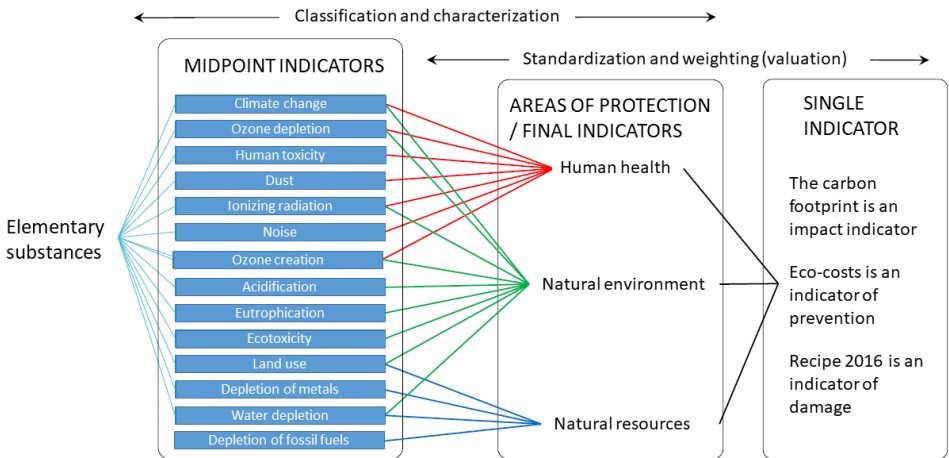

**Figure 2.** Assessment of environmental impacts by intermediate ("midpoint"), final and unique indicators for LCA study [14].

LCA generally aims to compare products or improve the environmental characteristics of a family of products. For energy production facilities, extensive LCA studies allow the impact of these complex systems to be assessed and it is possible to compare energy sources with each other. Table 1 classifies energy sources using the final carbon footprint indicator in grams of $CO_2$ equivalent per kilowatt.hour [15,16]. Renewable energies are low-carbon energies, they are about 10 to 100 times less emitting greenhouse gases than fossil fuels, except nuclear whose environmental impacts are of another nature.

**Table 1.** Emissions in grams of $CO_2$ equivalent per kilowatt.hour for different energy sources over their life cycle (median figures) [15,16].

| Type of Energy Source | eq. g $CO_2$/kWh | Type of Energy Source | eq. g $CO_2$/kWh |
|---|---|---|---|
| Coal | 820 | Large hydroelectricity | 24 |
| Diesel | 772 | Energies of the seas | 17 |
| Gas—Combined cycle | 490 | Offshore wind production | 12 |
| Biomass | 230 | Nuclear | 12 |
| Solar photovoltaic | 45 | Onshore wind production | 11 |
| Géothermie | 38 | Small hydroelectricity | 9 |

LCA can also be used to draft an environmental product declaration (EPD) on goods that are less complex than a complete energy production system. For example, Table 2 illustrates an EPD based on the LCA of an ABB electric machine (standardized results for a 1 kW machine) [17]. For a long lifespan

(25 years here), the environmental impacts are essentially those of the usage phase where the machine consumes electricity, which is logical for prolonged use such as this one.

**Table 2.** Environmental product declaration for a 250 kW ABB machine (HXR 355) [17]. (**a**) Inventory of materials. (**b**) Energy consumption and life cycle losses. (**c**) Environmental impacts on the manufacturing and use phases.

(**a**)

| Type of Material | kg/Product | kg/kW |
|---|---|---|
| Electrical Steel | 1252 | 5.01 |
| Other steel | 361 | 1.44 |
| Cast iron | 767 | 3.07 |
| Aluminum | 1.6 | 0.01 |
| Copper | 319 | 1.28 |
| Insulation material | 11 | 0.04 |
| Wooden packing material | 50 | 0.20 |
| Impregnation resin | 16 | 0.06 |
| Paint | 11 | 0.04 |

(**b**)

| Energy Form | kWh/Product | | | kWh/kW | | |
|---|---|---|---|---|---|---|
| | Manufacturing Phase | Usage Phase | Disposal Phase | Manufacturing Phase | Usage Phase | Disposal Phase |
| Electrical energy | 2087 | 2,003,541 | 136 | 8.35 | 8014.16 | 0.54 |
| Heat energy | 780 | - | - | 3.12 | - | - |

(**c**)

| Environmental Effect | Equivalent Unit | Manufacturing Phase | Usage Phase | Total Lifecycle |
|---|---|---|---|---|
| Global warming potential, GWP | kg $CO_2$/kW | 56.66 | 4028.82 | 4062.96 |
| Acidification potentiel, AP | kmol $H^+$/kW | 0.01 | 0.80 | 0.80 |
| Eutrophication | kg $O_2$/kW | 1.41 | 50.50 | 51.73 |
| Ozone depletion potential, ODP | kg CFC-11/kW | 0.00 | 0.00 | 0.00 |
| Photochemical oxidants, POCP | kg ethylene/kW | 0.05 | 0.92 | 0.97 |

Finally, the environmental sustainability of a product can be assessed globally by LCA and locally by impact studies on environments close to the installation. These studies are very specific to the production installation, carried out upstream of each project in order to obtain the authorizations of the State services and of the protection of the environment. Depending on the systems, the impacts analyzed relate to: direct emissions to air, water, soil, land and water use, noise, visual impact, impacts on wildlife and flora [18].

*2.4. Social Sustainability: Impacts on the Human Environment and Local Development*

Social sustainability corresponds to the impacts on the human environment of the electricity production installation: the advantages and disadvantages of the energy system with residents, businesses, decision-makers, managers and therefore on the various stakeholders. The main aspects studied are: local social development through the electrical service or the installation itself, the integration of the local population into the project and social acceptability, the client-manager relationship, etc. The challenge is to measure qualitatively and quantitatively the impacts on governance, education, health, safety and comfort while paying attention to the equitable distribution of benefits on the territory [10,11].

The transition from centralized energy production to spatially distributed production has revealed the importance of social acceptance. Indeed, public protests against renewable energy projects are proving powerful enough to slow or stop projects, like the "Not In My Back-Yard!" movement (NIMBY) Which has seriously scratched the onshore wind sector among others. The project developers have

understood this and are looking to integrate the contentment of the population upstream. It is possible to use social acceptance to discretize the maritime study area according to its potential for reduce conflict. For instance, in [6,19], the social criterion is considered from the very start of the reflections on the tidal farm alongside forecasts of produced energy and profitability. More generally, solutions considered optimal may in reality have low social sustainability. Theorized for energy distribution systems [20], the "socio-technical optimality gap" indicates that a classical conception can lead "to a solution that is apparently technically but not socially optimal".

It is clear that the social criterion prevails as the population is closely linked to the installation. This assumption comes to a head in the case of rural electrification systems in remote areas of developing countries. The effects on the human environment should be assessed as a priority when it comes to "energy for development". Development aid institutions and many researchers have focused on assessing the overall sustainability of electrification projects. In particular, Elisabeth Ilskog proposed a method for evaluating five dimensions of sustainability (technical, economic, environmental, social and organizational) with associated key variables and selected sustainability indicators for each key variable [11]. This evaluation is shown schematically in the diagram in Figure 3.

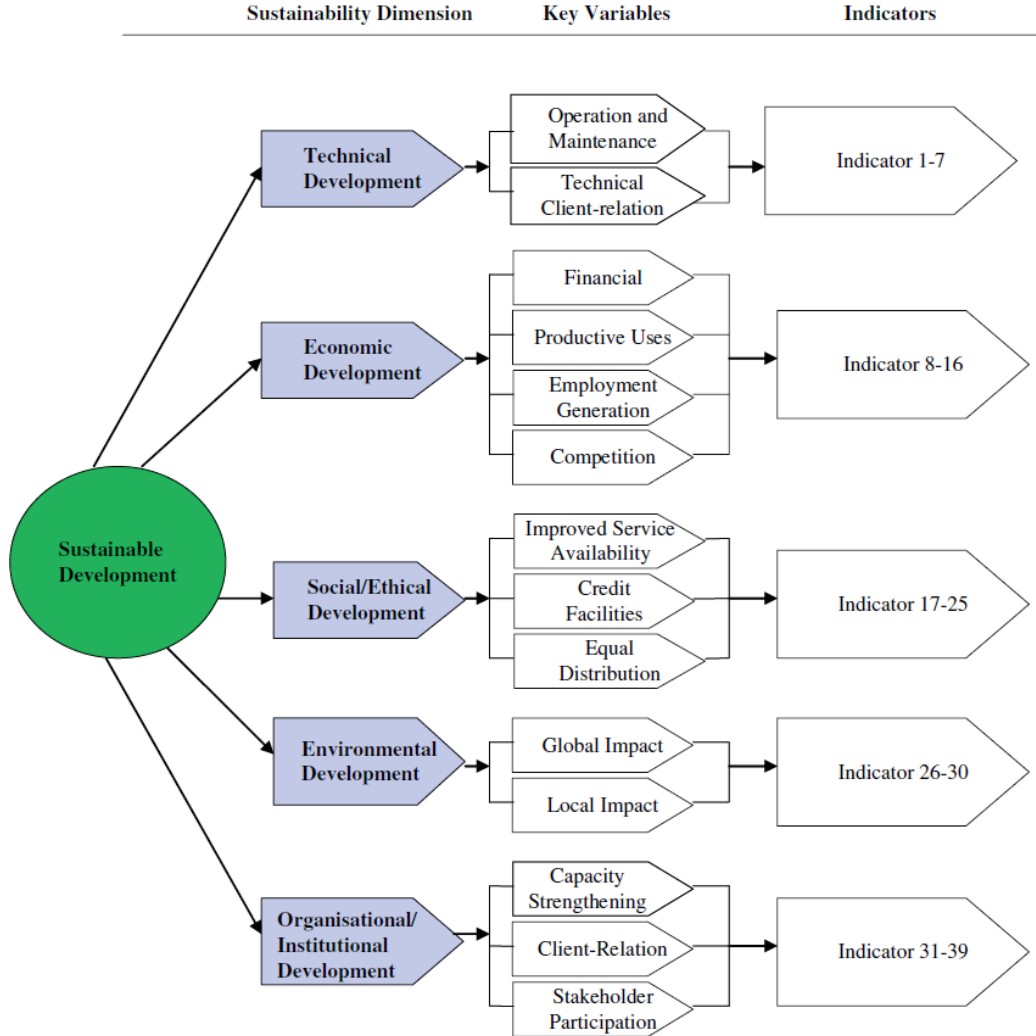

**Figure 3.** Assessment of the sustainability of rural electrification projects: dimensions, key variables and sustainability indicators [11].

Data is collected via operational reports or surveys and then compared to target values to normalize the results, so all indicators are quantitative even if they deal with quality. The choice

of indicators is important because they must be clearly defined, simple to understand and apply, transparent and fair. However, Ilskog warns about his method which cannot cover all the diversity of contexts by itself, it is necessary to add specific indicators to certain projects, it should rather be seen as a tool for decision-makers, a discussion support allowing to understand the impacts of an electrification project on the sustainable development of a community.

This multi-criteria assessment, like many others, assesses an energy access system after it has been used in relation to consumers. It would be interesting to note the impacts on other stakeholders by adding a reflection on the life cycle of the installation. Finally, it would be interesting to use this type of multi-criteria method not in evaluation during use but well before, from the design of the project. Upstream, this decision-making tool would have a lot of potential to inform discussions between all stakeholders on the sustainability of an electrification project, for example to choose one energy system rather than another according to its global impacts. In the following sections, we will therefore discuss this tool with an upstream use of projects in mind.

## 3. Imagine Solutions

After a detailed analysis of the need, while taking in the best of the context, comes the stage where solutions must be imagined responding to the identified problem. A product can be viewed from two complementary angles: either as a value proposition in relation to the need that has been identified, or as a technical solution for which we must think about the design and the associated processes.

### 3.1. Value Proposition

The first angle was hitherto addressed by sales representatives responsible for designing the business model of the product or service. The tool commonly used for this decisive step is the business model canvas presented by Osterwalder and Pigneur in [21] and shown in Figure 4. This table with nine entries makes it possible to build the value proposition in relation to the customer by thinking of the key parameters (partners, activities and resources), the customer relationship and the distribution channels against the background of the financial balance (expenses and profits). Based on this, the product is well thought out from the economic point of view but only this one. Aware of this bias and its consequences, Joyce and Paquin proposed in 2016 an enhanced version called "Triple Layered Business Mandel Canvas" by integrating the environmental and social perspectives of the product [22]. Figure 4b,c then serve as a support for thinking about environmental sustainability with the impacts on the life cycle (resources, manufacturing, distribution, use and end of life) and social sustainability with the social impacts of the product on all stakeholders (user, local communities, governance, employees). By adopting a reflection by board and between boards, the economic, environmental and social values of the product are well defined.

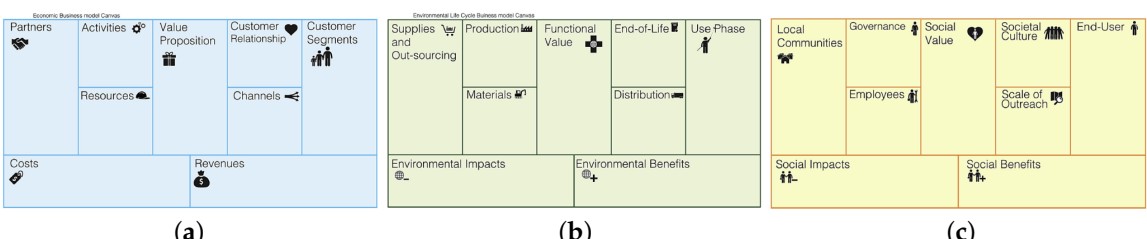

**Figure 4.** Triple Layered Business Model Canvas. (**a**) Business model. (**b**) Environmental lifecycle model. (**c**) Social model with stakeholders [22].

### 3.2. Design Process

Now, we look at the solution from the other angle, with the vision of the product designer who will imagine the design as well as the processes over the entire life cycle. The design processes have evolved over the last thirty years: from a linear or sequential design with an expert and then the other

to an integrated design where the experts collectively define all the formulations of the need in a global manner (in the 1990s). The integrated design mixes so-called internal actors (designers, engineers, financial, commercial and so on) with other so-called external (customers, suppliers, associations and so on) in order to find the compromise that best meets the requirements of all these actors [14]. In the 2000s, the design broadened its vision by understanding the environmental impacts on the entire life cycle of the product, we speak of ecodesign. This incorporates the ambition of environmental sustainability upstream of the project thanks to LCA. This reflection exists specifically on maintenance with the concept of eco-maintenance: optimizing the life of the installation by minimizing the environmental impacts caused by maintenance. An example of eco-maintenance by boat in offshore wind farms can be found in [23]. Finally, since around 2015, the notion of circularity completes the design process not only from an environmental point of view but also from an economic and social point of view: the aim is to extend the overall lifespan of resources thanks to reuse, reconditioning, recycling where waste again serves as raw materials.

Today, the design processes focus concretely on the environmental sustainability of the solutions and offer tools on the product design strategy. The graphic card in Figure 5 serves as an eco-design guide for creating or improving products by integrating the phenomena induced by circularity [14]. Knowing that it is too ambitious to consider all the criteria simultaneously, it is necessary to privilege the main aspects of the card in the case of the product considered as objectives of the updated specifications.

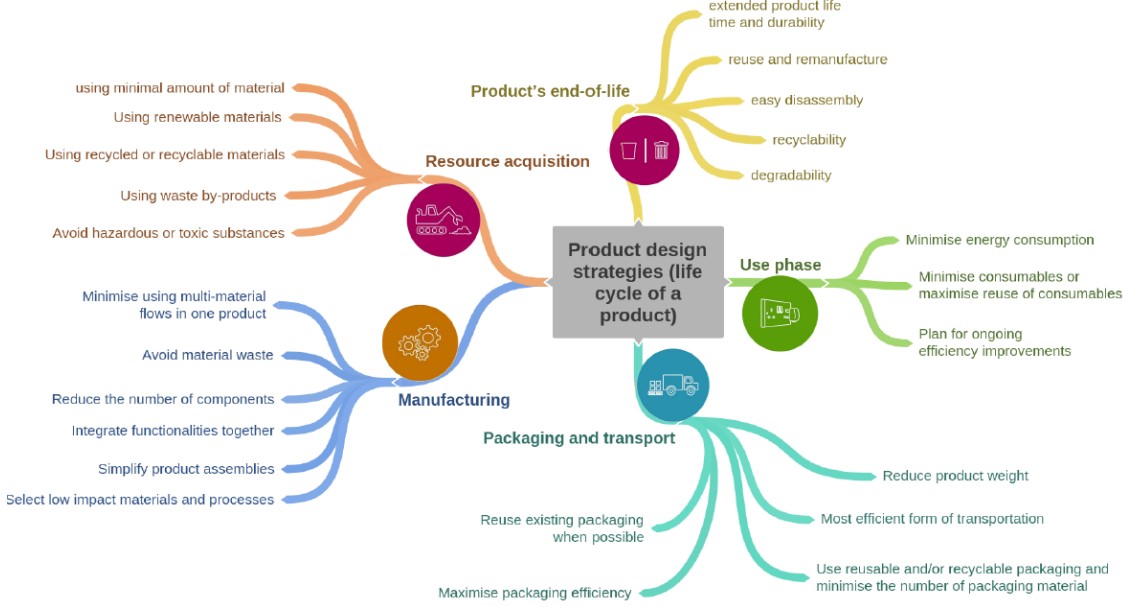

**Figure 5.** Product design strategy based on life cycle stages [14].

In addition, the so-called low and intermediate technologies for "low-tech" and "appropriate technology" are particular categories of technology which can be considered as alternating currents in the design process [24–26]. These philosophies integrate humans into the design, aiming for sustainability through the simplicity, robustness and resilience of the systems. Their value proposition tends to maximize the positive social impacts on different stakeholders by providing useful, sustainable and accessible technologies for the developing community. According to [25], low-tech is defined by the following qualifiers:

- Sustainable:

    - *Over time:* robust, modular, repairable, functional;
    - *By the search for sobriety:* efficiency up to the just necessary energy, economic burden and use of materials therefore optimizing the performance with respect to economic, human and environmental costs;
    - *From a societal point of view:* promoting solutions, activities and relocated knowledge, thereby reducing their environmental and social footprint and leaving the logic of exploitation;

- Accessible to the greatest number:

    - *Economically:* simple, optimized, robust, locally manufactured and repaired;
    - *In terms of knowledge:* democratize technique, open knowledge, creativity and ingenuity by orienting toward people in their environment.

By comparing the previous attributes with the map in Figure 5, we find that the low-tech philosophy goes in the same direction as circular ecodesign and even that it goes beyond with societal aspects. Thus, "low technologies" open innovative paths, interesting enough for the design of many products even to provide alternatives to high-techs [24].

Regardless of the design process, it always leads to the development of product specifications. In reality, this process is not entirely linear: the specifications are refined as the project progresses. At the beginning, the problem is not yet well defined and there are many uncertainties on the specifications. "Subsequently, as the project develops, the formulation of the problem evolves in coordination with the solutions proposed, and it is only shortly before the end of the project that the specifications become fixed and well defined" [27]. This is the "decision-information" paradox for design: the ability to make decisions on the project is inversely proportional to the knowledge of the product. This has an impact on costs: if the first phases of the process represent only 5% of the cost of the design activity, the choices still impact and fix 75% of the costs incurred over the lifetime of the product [27]. It is therefore important not to neglect these phases of analysis of the need of the solution because they are the most determining.

In the sketching phase, the approach adopted is rather systemic, "only the main criteria, constraints and objectives are accessible and definable" [27]. The level of competence required is at its maximum because the expert is enormously requested during this phase where 75 to 80% of the decisions are taken. At this stage, if it is possible to formulate an optimization problem, it would be complex (many criteria of distinct domains) with simple coarse models [28]. However, in the design phase, the approach is focused on the component and the simpler optimization problem based on complex detailed models.

## 4. Method

After defining the sustainability of an energy system in Section 2 and having exposed the concepts of value proposition and design process to imagine solutions in Section 3, we now link all these developments to propose a sustainable design method of electrical energy production systems. As a decision support tool, the method is used in the upstream phases of the project, during the design process. Thanks to a global vision, due to its multi-criteria nature, our method fosters the imagination of stakeholders who can thus outline a sustainable solution for the electricity production system. Thanks to the numerous quantitative and qualitative criteria of the four areas of sustainability, the systemic approach makes it possible to draw up the specifications produced as and when the discussions take place. This tool can be adapted to each project: depending on the specifics of the contexts, an adjustable weight system can be implemented to put the importance of each criterion into perspective [12]. Since everything is thought of from the design phase, the chances of success of the project increase and the user is more likely to obtain a fully sustainable solution that perfectly matches their needs.

We anchor the present method in the particular context of rural electrification and the first access to energy in isolated regions. As the development of the local community is a priority in these projects, we treat social sustainability first and we will see how a social requirement can affect other areas of sustainability. As this tool is used to make discussions progress, the method is refined as and when exchanges are made with cyclical questioning: the criterion of one domain can impact the criterion of another domain which can in turn have an effect on the first criterion or even another and so on. In Figure 6, we provide the example of looping on the local approach for manufacturing and maintaining the system. This environmental requirement has social and economic impacts on the community: creation of local jobs, reduction of expenses, direct relationship to technology promoting ownership and acceptance. These positive impacts promote social progress and community development. On the other hand, this requirement creates a new technical design constraint: components that are simple to manufacture and maintain, easy local access to repair materials. The designers must then limit the complexity of the solution which must be adapted to the level of knowledge of the community maintaining the system and to the level of skills of the local (regional, national) companies manufacturing the device.

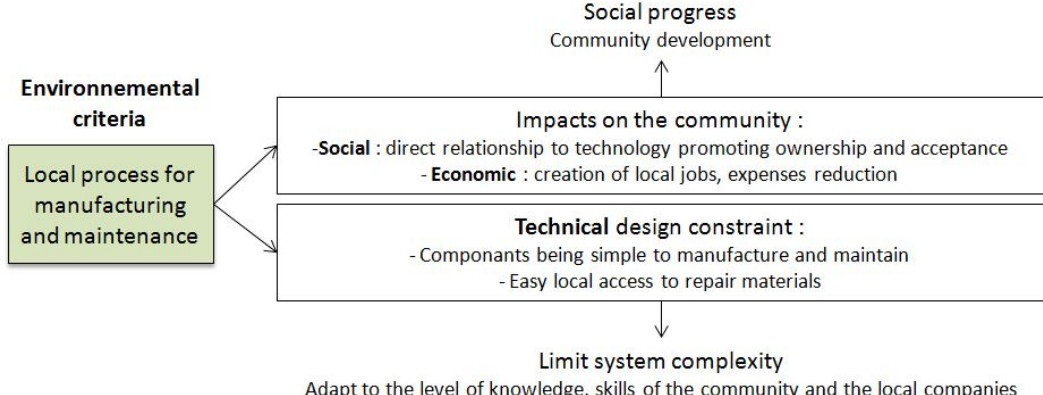

**Figure 6.** Completion of the local process for manufacturing and maintenance.

By way of illustration, a directive from the Indonesian Ministry of Industry has imposed since 2012 the local manufacture of electrical systems in order to improve the capacities of local industries in terms of renewable energy technologies [29]. The measure contributed to the development of a manufacturing capacity for micro hydroelectric plants which not only meets national needs but which also includes the export of components. Hydroelectric mini-grids must therefore have 64% local components for power plants with a power less than or equal to 15 MW.

As the method evolves over iterations, we propose here a version of the tool relevant to sketch an energy production system solution within the framework of a first access to energy for an isolated community. Each area of sustainability is broken down into key variables which group together indicators with a comment on their assessment. The bibliographic sources used are mentioned for each criterion. Finally, we opted for a life cycle vision in order to integrate the maximum impact into the reflections.

### 4.1. Social Criteria

The electrical energy production system serves above all the social and human development of the community on various levels such as education, health, well-being, personal and collective development. Table 3 presents the criteria of a rural electrification project from the point of view of social sustainability with the most relevant qualitative and quantitative criteria.

Social progress is conditioned by the new uses allowed by the energy system (lighting, ventilation, entertainment, productive uses, etc.), the time saved which can be devoted to education or leisure and

the positive effects on health and safety. Thus, the social requirements have mainly consequences on the electrical energy demanded by the community so they have repercussions on the technical and economic aspects, in relation with the sizing of the installation.

On the other hand, the links to be built between the system and the people, as a user, maintainer or manager, modify the technical specifications so as to obtain an accessible and adaptable solution in the low-tech philosophy. This situation allows a certain social emancipation through training with the feeling that the population realizes most of the action itself. Intrinsically the electricity production system allows more autonomy than a connection to the network.

**Table 3.** Social criteria of the sustainable design method for a rural electrification project.

| Key Variable | Indicator | Evaluation | Source |
|---|---|---|---|
| Community development | Electrified health centers and schools | % of electrified public utility spaces | [11,30] |
| | Street lighting | Number of lamps every 40 m | [11,30] |
| | Places with TV access, internet | % of public and private places | [10,11] |
| | Share of the electrified population | % of the total population | [11,30] |
| | Electrical appliances | Number of household electrical appliances | N.A. |
| | Leisure time | Leisure time | [10,31] |
| Integration of the population | Commitment of the community, social acceptance | Investment in time, participation in the project, interest, curiosity, appropriation of the solution | [11,12,31] |
| Capacity strenghtening | Share of the population working for the service | % of local employees for the installation/maintenance of the service | [11] |
| | Share of staff with appropriate education | % of staff with technical training | [10,11,30] |
| Customer relationship | Satisfaction | Level of satisfaction with the service | [10,11,30] |
| Health safety | Health/safety of employees, users | Accidents, electrical incidents, health effects | [10,11,18,30] |
| Education | Studies at home, at school | Number of hours of study for children | [10] |

Some of the social criteria in Table 3 seem difficult to foresee and to take into account within the design stage. We believe that removing them from the method under this pretext would be detrimental to the objective of sustainability because dialogue between all the stakeholders upstream is all the more important for social sustainability than for the rest. Developers and especially designers of solutions must grasp these social criteria, rework them according to the local context and draw guidance on technologies even if it means transforming them into criteria in other dimensions, or purely quantitative ones, for the comparison of solutions, otherwise viable.

## 4.2. Technical Criteria

The technical sustainability criteria are linked to the performance of the production system as well as its durability over time, its accessibility in terms of knowledge and means. The technical criteria are arranged by phase of life, from manufacturing to end of life. Table 4 presents the technical aspect of the sustainable design method.

Over the entire life cycle, the desire to anchor the territory for the provenance of materials and the relocation of activities (manufacturing, installation, maintenance, end of life) pursues the

objective of autonomy and emancipation through a low-tech philosophy [25]. This presupposes a perfect knowledge of local activities, and if they do not exist, it is necessary to judge the possibility of developing them and of producing local expertise and know-how.

The technical performance in operation (lifetime, power, energy, efficiency, etc.) can all be taken into account by the technical expert in the specifications of the installation. Compatibility with the future network and national standards are very important compared to a possible connection with the national electricity grid, failing which the electrical installation will become obsolete. The quantity of oil avoided is used to compare the system with production based on a diesel generator. If the production is based on renewable energy (without fuel), the yield is not so important because the primary energy is free: it is especially necessary to maximize the ratio "produced energy" on "installation costs" (therefore the LCOE) and more broadly, increase the energy return on investment (EROI). By taking up the attribute of low-tech [25], the objective is to strive for an optimization of technical performance with regard to economic, human and environmental costs.

**Table 4.** Technical criteria of the sustainable design method for a rural electrification project.

| Life Steps | Indicator | Evaluation | Source |
|---|---|---|---|
| Manufacturing | Share of elements manufactured in the country | Knowledge of the country's industries, ease of access to materials, level of complexity | N.A. |
| Installation | Ease of installation | Material and human resources required, level of complexity | [31] |
| Operation | Installed power | Average and peak power in kW | N.A. |
| | Electricity produced | kWh/year | [11,12,31] |
| | Energy efficiency | % primary energy converted into electricity | [11] |
| | Availability | % hours/year | [11] |
| | Load factor | % energy produced/producible | [11,30] |
| | Technical losses | Electrical losses between the end of production and the consumer | [11] |
| | Operating hours/day | | [11,30] |
| | Uses permitted by the service | Lighting, ventilation, etc. | [10] |
| | Compliance with country standards | Yes/no | [11] |
| | Compatibility with the future network service | Yes/no | [11,30] |
| | Service life of system elements | N.A. | |
| | Quantity of oil avoided | Tonnes of equivalent oil avoided/year (diesel generator case) | [12] |
| Maintenance | Reliability, security, robustness | Element failure rate, risk of accident, stability of the micro-network | [12,18,31] |
| | Ease of maintenance | Material and human resources required, level of complexity | [11,31] |
| | Availability of spare parts | Share from the community, the country; access to materials | N.A. |
| End of life | Ease to recycle or reuse | Locally, in the country | N.A. |

Concretely, the ease of installation and maintenance are linked to the complexity of the system but also to the masses and volumes involved. For the most constrained areas, it is preferable that

the system be dismantled into manipulable, interchangeable sub-parts and transportable safely by a few people.

We saw in Section 2 that high availability was partly enabled by reliable systems reducing the duration and the occurrence of failures [5]. The method mentions this aspect of sustainability so that discussions can focus as soon as possible on a choice of technologies, components and materials that are reliable, robust and/or easily repairable.

### 4.3. Economic Criteria

The criteria of economic sustainability introduce the economic performance of the production system for managers and financial accessibility with the price of electrical service for users. The method also presents the effects of the electrification project on the community of users and the local economic environment. Table 5 presents the economic aspect of the sustainable design method for the electricity production system.

**Table 5.** Economic criteria of the sustainable design method for a rural electrification project.

| Key Variable | Indicator | Evaluation | Source |
|---|---|---|---|
| Managerial financial perspective | Return on investment, payback time | Number of years | [12] |
| | Net present value of investment | Price of the electricity generation solution | [12] |
| | CAPEX | €/kW | [11,12,31] |
| | OPEX | €/kWh | [11,12,31] |
| | LCOE | €/kWh | N.A. |
| | Profitability | Profit on the sale of electricity/total bills (1 year) | [11] |
| | Share of reinvested profits | Profit on the sale of electricity/total material depreciation cost (1 year) | [11] |
| Job creation | Share of electricity used for income-generating activities | % of electricity produced (by individuals and businesses) | [11,31] |
| | Number of companies, jobs involved in manufacturing, maintaining the service | Local Business (share or number); Service jobs: salary and benefits compared to other companies | [10,12] |
| Economic environment | Possibilities of microloans to connect to electricity | Number of microloans | [10,11,30] |
| | Subsidies for the service | of the government, of NGOs for the inhabitants, the electrical manager | [11,30,31] |
| | Compatibility of the solution with economic activities | Impacts on agriculture, fishing, tourism ... | [12,18] |
| Household budget | Price of electricity | €/kWh or €/service, use | [10] |
| | Productive time | Overtime to work with lighting | [10] |

From the manager's point of view, financial performance is always present and well estimated upstream by financial experts. The CAPEX, OPEX and LCOE costs are linked to the power of the installation and other technical performances, visible on Table 4. Costs are reduced if the installation is smaller, less complex and if the manufacturing and maintenance activities are local.

From the point of view of the subscribers, the budget devoted to the electrical service must be aligned with the old energy expenditure of the home. Similarly, micro-credits must be in accordance

with the financing plan: if the household spent €5/month on kerosene, monthly payments should be around €5/month for electrical service.

The electrical installation is beneficial to the local economic environment by stimulating activities, creating jobs directly for service and indirectly for individuals and electrified businesses. The effect is all the stronger when the quantity of energy available is large and the power high to allow high-level electrical uses on the energy access scale (refer to Table 3). However, the electrification project can compete with activities of an economic nature already or soon to be established in the territory. Therefore, a stakeholder arbitration is necessary in order to agree on an optimum.

### 4.4. Environmental Criteria

The environmental sustainability criteria are linked to the impact of the production system on the environment on a global and local scales over the entire life cycle. Table 6 presents the environmental impacts to be anticipated in the sustainable design method.

**Table 6.** Environmental criteria of the sustainable design method for a rural electrification project.

| Scale | Indicator | Evaluation | Source |
|---|---|---|---|
| Over all impact | Share of renewable energy in electricity production | % kWh produced | [11,30] |
| | LCA of the system | On all the elements | [31] |
| | $CO_2$ reduction potential | Tonnes of $CO_2$ avoided/year (diesel generator) | [11,12,30] |
| Local impact | Land, water use | $m^2$, $m^3$ | [12,18,31] |
| | Noise level | Incremental noise level dB added * number of people affected; in all environment | [12,18] |
| | Aesthetic visual impact | On the landscape, the heritage | [12,18] |
| | Compatibility with non-economic activities | Leisure | [18] |
| | Electric network and access road | km from network, road for service (erosion) | [12] |
| | Emissions to water, air, soil | Contaminations, modifications; pollutants, gases | [18] |
| | Temperature change | In all environments | [18] |
| | Terrestrial/aquatic fauna/flora | Movement of animals, fish, others; aquatic, terrestrial vegetation | [18] |
| | Extreme climatic condition | Disturbances of the system (natural causes) | [11,30] |

The environmental criteria are varied, relevant on local and global scales, and on the short and long term. They concern natural environments (air, water, soil) and living species (humans, fauna or flora).

Global and local impacts must be considered before the electrification project. If the effects appear to be too significant, the developers have the choice between making structural modifications to the imagined solution or opting for another less harmful solution. LCA especially makes sense for the comparison between two solutions, this is also the case for all the criteria in Table 6.

## 5. Discussion on the Method

The sustainable design method that we offer consists of the four tables Tables 3–6 for the technical, economic, environmental and social sets of criteria. These tables bring together all the criteria necessary

to think about sustainability upstream of rural electrification projects. They constitute a support for discussion within the framework of an integrated design where all the stakeholders intervene (developers, community, manager, local policy, manufacturers).

The method may seem complex and difficult to use because it includes many criteria which are not obvious to assess in the upstream phase. However, all these sustainability criteria must appear at the earliest in the project. We believe that the stakeholders should discuss each of these criteria during the preliminary stages of the project, when moving from the clear issue to the imagination of solutions. It is precisely in this cloud of ideas where decisions are decisive for the continuation of the project that the lasting solution takes shape. By opting for a less complete vision, the project is likely to stray from a sustainable trajectory and it will be too late or too expensive to correct itself in the following phases or once the system is delivered.

At the end of the design stage, the chosen solution is the result of debates, of arbitration between several opposing criteria. In general, economic criteria significantly orient the solution towards a reduction in technical, environmental and social performances. Our multi-criteria method with its four tables gives the same importance to the four areas of sustainability. Thus, discussions between all stakeholders are balanced and the center of gravity of the solution is not shifted to economic concerns. Then, it is necessary to question the way in which the actors interested in the sustainable approach ultimately choose the solutions. How much can companies turn to a better solution on $CO_2$ emissions that does not improve the LCOE? How to offer a complete electrification when the constraint of profitability poses a selling price only bearable by 30 to 50% of the population? We do not have answers to these delicate questions but our method allows all the interlocutors to discuss the possible solutions. They are then free to find or create financial or regulatory tools to leverage and support their choices.

It is possible to find, in the literature, methods combining two to three criteria (technical and economic mostly, environmental sometimes) in design studies [32,33]. Defining and applying social criteria requires generally compromises to quantify some aspects that otherwise would not be taken into account [20]. This creates the need for dedicated methods [22] from which the one presented in this paper derives, associating all criteria in a sustainable perspective. The difficulty relies in the end in the capacity to quantify indicators that would otherwise not be taken into account in an optimization and design/sizing tool.

In practice, the sustainable design method could be implemented in a multi-objective optimization tool after having translated all the criteria into simple mathematical models. The formulation of the problem would be rather complex because of the difficulties in making the qualitative criteria quantitative and in comparing the criteria between them. Within each domain, it is possible to assign importance weights to each criterion in accordance with the requirements of the stakeholders and the context. Then we would still have to set the weights of each area equally or with differences. Finally, an additional difficulty would be to link these different criteria to the optimization variables, often dimension or control variables, naturally linked to the technical criteria, but much less to the others. Optimization tools will find it difficult to find a set of optimal multi-criteria solutions if certain objectives have little influence on the solutions, while they are moreover very sensitive to others (without going into the details of continuous and binary variables).

## 6. Conclusions

After analyzing the way to assess the sustainability of energy systems in operation, we proposed a multi-criteria design method to understand their sustainability upstream of the project, aiming at the development of energizing isolated local community for the first time without the support of a national grid. Presented in the form of four tables of criteria for the technical, economic, environmental and social fields, the method serves as a support for discussion within the framework of an integrated design where all the stakeholders are invited to take part (developers, community, manager, policy local, manufacturers).

The next step would be to apply our sustainable design method to a practical case of the problem of access to energy of isolated areas. We should then take the same path to imagine the technical solution.

**Author Contributions:** Conceptualization, K.M., L.G. and V.D.; methodology: K.M., L.G. and V.D.; software: L.G.; validation: K.M., investigation, K.M.; resources: L.G.; writing—original draft preparation: K.M.; writing—review and editing: K.M. and V.D.; visualization: K.M.; supervision: L.G. and V.D.; project administration: L.G. and V.D. All authors have read and agreed to the published version of the manuscript.

**Funding:** This research received no external funding.

**Conflicts of Interest:** The authors declare no conflict of interest.

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
