# Peer review of "Multi-Criteria Method for Sustainable Design of Energy Conversion Systems"

_sustainability, doi:10.3390/su12166513_

Round 1
Reviewer 1 Report
The article looks fine. From a general perspective, I think the discussion section could be expanded to indicate the difference between the proposed and the existing solutions.
Author Response
Please refer to the uploaded pdf with the responses.
Regards,
Vincent Debusschere, on behalf of the authors

Reviewer 2 Report
The article title should incorporate energy production to better identify the topic (e.g. "Multi-criteria method for sustainable energy production solution design")
I would recommend a sentence at the beginning of the abstract to introduce the focus of the article (i.e. the need to develop sustainable energy production solution designs for isolated communities lacking national energy grids).
The Introduction section should begin with one paragraph introducing the article topic, and the background and importance of the topic.
Author Response

(The authors gave the same response as above.)

Reviewer 3 Report
The authors mentioned that sustainability has technical, economic, environmental and social angles so it i better to divide the sections in the paper based on those titles and then consider its design and solutions.
The introduction is short and needs to be expanded and developed by updated references and in methodological approach.
It is interesting to provide some successful real case studies on each criteria for the provided energy systems.
Author Response

(The authors gave the same response as above.)
